# Motion of dust particles in dry snow under temperature gradient metamorphism

Pascal Hagenmuller[1], Frederic Flin[1], Marie Dumont[1], François Tuzet[1], Isabel Peinke[1], Philippe Lapalus[1], Anne Dufour[1], Jacques Roulle[1], Laurent Pézard[1], Didier Voisin[2], Edward Ando[3], Sabine Rolland du Roscoat[3], and Pascal Charrier[3]

[1]Univ. Grenoble Alpes, Université de Toulouse, Météo-France, CNRS, CNRM, Centre d'Études de la Neige, Grenoble, France
[2]Univ. Grenoble Alpes, CNRS, IRD, Grenoble INP, IGE, F-38000 Grenoble, France
[3]Univ. Grenoble Alpes, CNRS, Grenoble INP, 3SR, F-38000 Grenoble, France

**Correspondence:** P. Hagenmuller (pascal.hagenmuller@meteo.fr)

**Abstract.** The deposition of light-absorbing particles (LAPs) such as mineral dust and black carbon on snow is responsible for a highly effective climate forcing, through darkening of the snow surface and associated feedbacks. The interplay between post-depositional snow transformation (metamorphism) and the dynamics of LAPs in snow remains largely unknown. We obtained time series of X-ray tomography images of dust-contaminated samples undergoing dry snow metamorphism around -2°C. They provide the first observational evidence that temperature gradient metamorphism induces dust particle motion in snow, while no movement is observed under isothermal conditions. Under temperature gradient metamorphism, dust particles can enter the ice matrix due to sublimation-condensation processes and spread down mainly by falling into the pore space. Overall, such motions might reduce the radiative impact of dust in snow, in particular in arctic regions where temperature gradient metamorphism prevails.

*Copyright statement.* TEXT

## 1 Introduction

Dust in snow is an important driver of snowpack evolution (Dumont et al., 2014; Ginot et al., 2014; Painter et al., 2018; Skiles et al., 2018). Mineral dust primarily affects the snow optical properties by lowering the albedo, which governs the amount of solar energy that is absorbed, in the visible wavelengths (Warren and Wiscombe, 1980). This albedo decrease leads to accelerated metamorphism and possibly faster melt rates (Flanner et al., 2007; Tuzet et al., 2017). Several snow covered regions frequently undergo major dust outbreaks leading to a deposition of large amounts of mineral dust in the snowpack, which significantly decreases the snow albedo by several percent (e.g. Di Mauro et al., 2015). Skiles and Painter (2017) measured dust concentration as high as several milligrams per gram after major dust outbreaks in Colorado (USA).

While several studies have quantified the radiative impact of dust on snow evolution (e.g. Skiles et al., 2012; Painter et al., 2012), large uncertainties remain. They pertain in particular to (i) uncertainties of the dust refractive index (due to the influence of its geometry and chemical composition) (Caponi et al., 2017), (ii) poor knowledge of the mixing state of the dust particles in the ice matrix (Flanner et al., 2012) and (iii) imperfect representation of the vertical distribution of dust in the snowpack and related impacts (Dumont et al., 2014; Tuzet et al., 2017). More specifically, when light absorbing particles (LAPs) are mixed internally within the ice matrix, their optical impact increases due to lensing effects, when compared to the case when they are mixed externally (Flanner et al., 2012). Indeed, Flanner et al. (2012) showed that black carbon (BC) absorption efficiency is increased by a factor of roughly two in the case of internal mixture compared to the external mixture case. Besides, for a given amount of LAP, the LAP vertical profile within the snowpack strongly impacts their radiative effect. Typically, the albedo decreases more when the mass of LAPs is concentrated in the first centimeter compared to the case when the mass is distributed over several centimeters (Aoki et al., 2000; Dumont et al., 2014).

Up to now, only a few studies have been dedicated to the impact of post-depositional snow transformation, i.e. snow metamorphism, on LAP location and to the non-radiative interplays between LAP and metamorphism. Voisin et al. (2012) and Doherty et al. (2010) reported lower BC content for dry snow layers which had undergone temperature gradient metamorphism as compared to other snow layers. These authors suggested that a self cleaning mechanism is active in these snow layers due to sublimation-condensation processes, which results in a reduction of the snow layer LAP content (Doherty et al., 2010). Trabant and Benson (1972) measured a lower electrical conductance of melted snow samples which had undergone strong temperature gradient metamorphism as compared to other snow samples, and also interpreted this observation as a purification process related to the formation of depth hoar. Meinander et al. (2014), Skiles and Painter (2017) and Seidel et al. (2016) provided some observational evidence of non-radiative impact of LAP in wet snow, namely changes in liquid water retention and grain size.

In this study, we introduce the first *in operando* observation of non-radiative interplays between dust and dry snow metamorphism using X-ray tomography. More specifically, we investigated the impact of isothermal and temperature gradient metamorphisms on dust particle location in snow.

## 2  Material and Methods

### 2.1  Experimental set-up

We acquired tomographic time series of two similar snow samples, containing a thin layer of mineral dust and undergoing metamorphism under isothermal conditions and moderate vertical temperature gradient conditions.

The mineral dust used to contaminate the snow samples was Mongolian sand. Its mineralogical composition was 60% Quartz, 20% Albite, 11% Calcite, 9% Illite and traces of Kaolinite ($\sim$0.1%), as measured by X-ray diffraction using the Rietveld technique (Nowak et al., 2018). Elemental concentration for major constituents of mineral dust was determined by wavelength-dispersive X-ray fluorescence following Formenti et al. (2010), resulting in Si/Al = 4.3 and Fe/Ca = 0.55. The total

dust mass was estimated from this elemental composition as in Caponi et al. (2017), leading to an iron mass fraction in the measured dust of $MR_{Fe\%} = 3.6\%$. These values are within the typical range of dust composition of Caponi et al. (2017).

The two samples composed of recent snow (DF/RG), according to the international classification for seasonal snow on the ground (Fierz et al., 2009), were prepared as follows. Fresh snow was first collected in the field and stored at -20°C for 9 days. It was then processed in a cold room at -5°C: a snow layer was sieved with a sieve size of 1.6 mm on a 40 x 60 cm polystyrene plate. Half of its top surface was contaminated by manually sieving dust with a metallic sieve (0.5 x 0.5 mm$^2$ holes). To smoothen this process, a textile mesh (0.5 x 1 mm$^2$ holes) has been previously folded and placed in the sieve before operation. Another snow layer of same properties was then sieved on top of the whole surface. Finally, this preparation was left sintering in a closed insulated box for 24 hours. Snow cores with a diameter of 10 mm were vertically extruded from the prepared snow so that only about half of the core horizontal cross-sections were contaminated with dust. The top and bottom of the cores were then cut, resulting in a dust surface located in the middle of the core with a total height slightly above 8 mm. The cores were then smoothly inserted into the aluminum sample holders (cylinders of diameter and height of 10 mm) and sealed with copper columns on which 1 mm thickness ice lenses have been previously grown. The snow samples thus contained a thin horizontal dust layer in their middle spreading only one half of their cross section (see Fig. A1). The samples were then stored at -20°C for 4 days.

In order to accurately control the temperature boundary conditions of a sample while it is scanned by X-ray tomography, we used a previously developed cryogenic cell (Calonne et al., 2015), which is based on a thermo-electrical regulation and a vacuum insulation. The temperature of the Peltier modules are monitored during the whole experiment and controlled with a relative precision of about ±0.01°C. The Pt100 probes were calibrated together in a thermo-regulated bath so that the accuracy of their temperature difference is below ±0.02°C and their absolute temperature is known at ±0.05°C. The insulation with vacuum is designed to avoid any inadvertent lateral heating of the sample (Calonne et al., 2015). One sample was subjected to isothermal conditions at -2°C. Due to an imperfect thermal connection in the experimental setup (later detected on X-ray images), a slight temperature gradient was unavoidable but estimated to be lower than 5 K m$^{-1}$ during this experiment. The other sample was subjected to a downward vertical temperature gradient of 19 K m$^{-1}$ (top is cold, bottom is warm) at a mean temperature of -2°C.

After complete setting of the boundary conditions, the snow samples were regularly scanned at a resolution of 7.5 $\mu$m via X-ray tomography (EasyTom XL Micro, RX Solutions) at Laboratoire 3SR. The X-ray tube was powered by a current of 100 $\mu$A and a voltage of 80 kV. Assuming a reasonable X-ray source efficiency of 0.1%, i.e. a very small fraction of the energy input is effectively converted into X-ray photon energy, it can be assumed that the absorption of the X-rays by the sample holder, the snow and the dust has no significant impact on the sample temperature. The detector was composed of 1920 x 1536 pixels with a physical pixel size of 127 $\mu$m. Each scan consisted of 1440 radiographs covering a 360° rotation. One radiograph was computed as the average of 5 frames captured at one frame per second. A complete scan lasted 2 h and was started every 3 h, on average. The isothermal experiment lasted about 100 h, leading to 32 tomographic images. The temperature experiment lasted about 200 h leading to 75 images. The 3D image resolution (7.5 $\mu$m) and the subsequent tomograph settings were chosen to capture most of the dust particles with a mean mass diameter around 5 $\mu$m (according to the Coulter analysis for dust particles

with a diameter between 0.6 and 20 $\mu$m, see next paragraph) while keeping the scanning time (2 h) short enough to correctly measure the microstructure time evolution. This tomographic data thus comprises the most resolved time series, 7.5 $\mu$m spatial resolution and 3 h average temporal resolution, of snow evolution ever measured.

At the start of the tomographic time series, according to the first tomographic image, the sample of the isothermal experiment was characterized by a density of 215 kg m$^{-3}$ and a specific surface area of 32 m$^2$ kg$^{-1}$, the sample of the temperature gradient experiment by a density of 230 kg m$^{-3}$ and a specific surface area of 32 m$^2$ kg$^{-1}$. A post-mortem Coulter analysis of the samples indicated that the dust content was about 0.5 mg g$^{-1}$ for dust particles with a diameter between 0.6 and 20 $\mu$m.

## 2.2 Image processing and analysis

The sets of radiographs were reconstructed into 3D images of the different attenuation coefficients of the materials composing the samples namely air, ice, aluminum sample holder and dust. Due to slight movements of the tomograph elements, the different images of one time series do not necessarily share the same coordinate system. The images were thus registered assuming that the sample holder does not move nor deform with time. To this end, the python package SimpleITK (Yaniv et al., 2018) was used to find the composition of an isotropic scaling and a rigid transformation that best matches the sample holder gray-scale image between the considered and the reference images, according to a gray-scale correlation metric. The sample holder was not perfectly cylindrical but contained different marks so that the seven degrees of freedom of the transformation can be unambiguously determined. Translations up to 50 $\mu$m were corrected with this procedure, while the other degrees of freedom remained mainly unchanged.

The segmentation of images containing multiple materials is notoriously difficult due to the presence of noise and partial volume effects increased by blur. Following the approach of Wolz et al. (2010), we performed a 4D (3D space + time) energy-based segmentation. The energy function was composed of a data term representing intensity priors, i.e. the likelihood of a voxel to be composed of one material according to its grayscale value, and a regularization term representing the cost for voxels to be segmented differently from their spatial and temporal neighboring voxels. The data term was based on Bayesian classification of material mixture, accounting for partial volume and gradient modeling (Bromiley and Thacker, 2008). The regularization term was set as the area of the 4D surface separating the different materials in space and time. Hagenmuller et al. (2013) showed that the regularization using the surface area between air and ice is of particular interest in the segmentation process by mimicking the curvature-driven metamorphism. Wolz et al. (2010) showed that regularization in time is essential to get a consistent segmentation of time series.

Three main geometrical characteristics were derived from the tomographic data: volumetric contents, surface areas and vertical displacements with time. To avoid edge effects due to the sample holder and the ice lenses, these characteristics were computed on a sub-volume with a 1 mm margin to the sample holder and to the lens initial position, i.e. on a centered cylinder with a diameter of 8 mm and a height of 6 mm. Volumetric contents were computed by voxel counting in the segmented images. Typical size of the dust particles, defined as non-connected dust parts, was computed as the radius of a sphere with the same volume. Surface areas (total area and contact area) were computed on the segmented images by the Crofton approach which showed good accuracy on anisotropic structures (Hagenmuller et al., 2016). An anisotropy factor $Q$ was also calculated

based on the ice-air stereological surface areas in the vertical direction $S_z$ and horizontal directions $S_x$ and $S_y$. It was defined as $Q = (2/S_z)/(1/S_x + 1/S_y)$ (Calonne et al., 2014). We also computed the ratio $\eta$ between the dust-ice contact area and the dust surface area for each particle. A high $\eta$ value means that the particle is almost completely embedded in the ice matrix. Due to the limited accuracy in computing surface areas of discretized objects, which are small compared to the voxel size, the

dust particles with a volume smaller than $5^3$ voxels were not taken into account here. Nevertheless, small contact areas of the order of the resolution between the ice matrix and the particles might still be missed due to segmentation errors. Computation of the displacements of the dust particles and the ice interface was more complex. Almost no dust particle entered or exited the considered volume during the experiments, the movement of the overall dust center of mass thus provided a robust estimation of the bulk dust movement. It was not possible to measure the total water fluxes with a similar approach since ice and vapor

entered and exited the measured volume. The movement of the ice interface between two successive scans was calculated using an implicit representation of the ice surface as the zero-isosurface of a certain level set function $\phi$. The function $\phi$ was here defined as the signed euclidean distance map of the segmented ice component, smoothed with a Gaussian kernel of standard deviation of 2 voxels. The levelset equation links the interface velocity $\mathbf{v}$ to the temporal and spatial derivatives of $\phi$: $\frac{\partial \phi}{\partial t} + \mathbf{v} \cdot \nabla \phi = 0$. The time derivatives $\partial/\partial t$ were computed between two consecutive images generally separated by

3 hours. The spatial derivatives $\cdot\nabla$ were computed by convolution with a first-order derivative operator. Assuming that $\mathbf{v}$ is normal to the interface, it was possible to recover the full velocity vector and, in particular, its vertical component. Lastly, each dust particle was tracked individually between two consecutive gray-scale images by volumetric digital image correlation (Hall et al., 2010; Yaniv et al., 2018). This technique captures sub-resolution displacement of dust particles but fails to track large dust displacements (e.g. exceeding the particle size). We rejected the displacements when the tracking metric, namely

normalized cross correlation, was lower than 0.9 or when they were computed on a too small particle (particle volume lower than $5^3$ voxels). Exemplary slices showing the binary segmentation and the ice interface velocity computation are provided in the appendix (Fig. A2).

## 3   Results and discussion

### 3.1   Snow and dust microstructure evolution captured by tomography

The snow under temperature gradient conditions evolved into faceted crystals and depth hoar (FC/DH, Fierz et al. (2009), Fig. 1a) while the snow under isothermal conditions remained recent snow (DF/RG, Fierz et al. (2009), Fig. 1b). Movies of the microstructure evolution are provided as supplementary information (Movies S1 and S2). The time evolution of the snow microstructural properties is shown in Fig. 2. Snow density remained almost constant with time, with a standard deviation of 0.7 kg m$^{-3}$ under temperature gradient conditions and 0.6 kg m$^{-3}$ under isothermal conditions (Fig. 2a). Temperature

gradient metamorphism creates an upward water vapor flux and an apparent downward solid ice flux (e.g. Yosida, 1955; Flin and Brzoska, 2008; Pinzer et al., 2012). The constant density under these conditions thus indicates that the water fluxes (vapor and solid) at the sample top and bottom were nearly equal, without any apparent edge effect on the measured snow volume. The specific surface area decreased in a linear fashion with time at a mean rate of -0.043 m$^2$ kg$^{-1}$ h$^{-1}$ under temperature

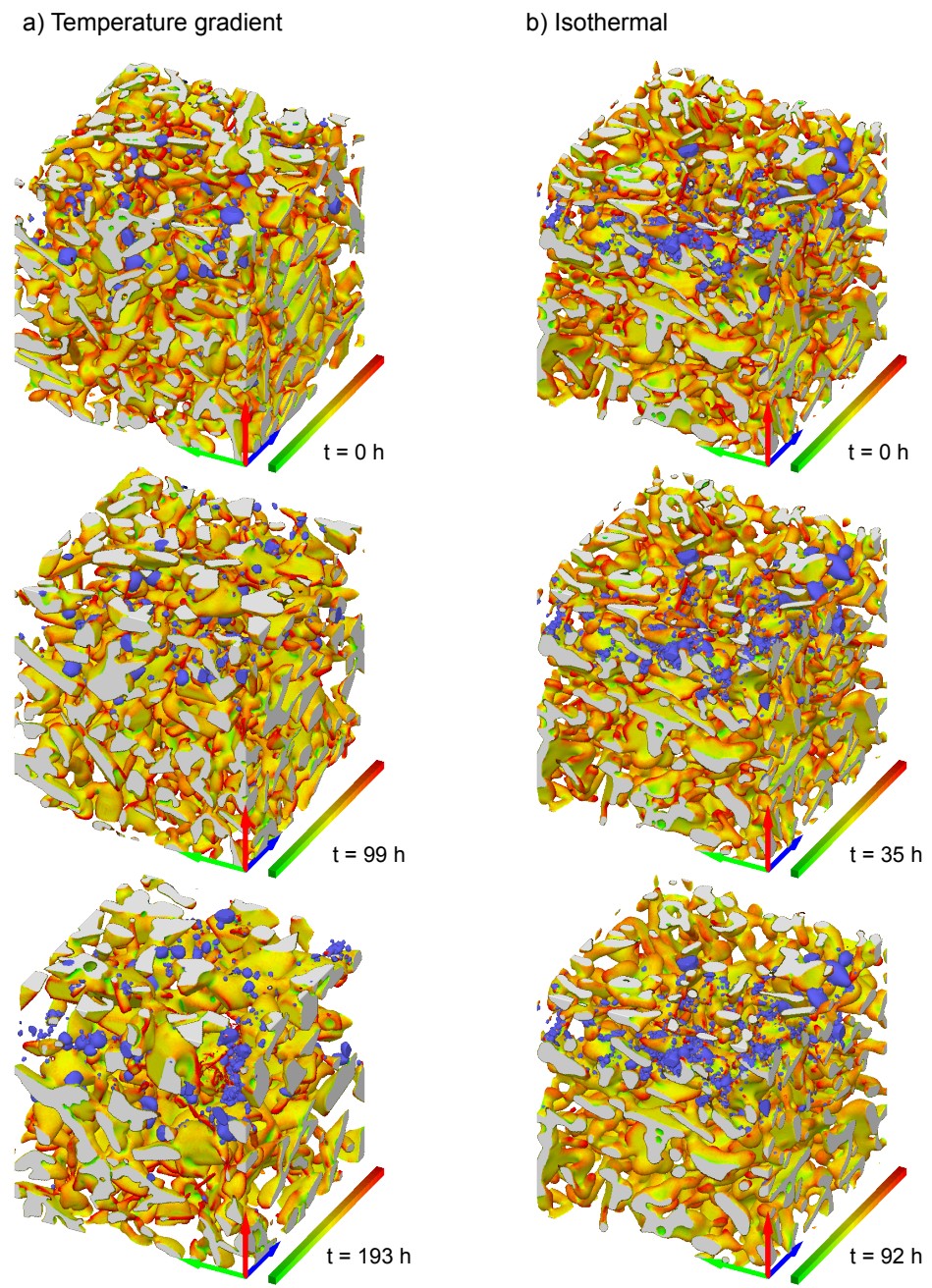

**Figure 1.** Microstructure of the samples at initial, intermediate and final stages. Only subvolumes of size $3^3$ mm$^3$ are shown with a vertical orientation corresponding to that of the experiment. The colormap represents the mean curvature from green (-25 mm$^{-1}$) to red (+25 mm$^{-1}$) and the colored bar is 3 mm long. The dust particles are shown in blue.

gradient conditions and -0.066 m$^2$ kg$^{-1}$ h$^{-1}$ under isothermal conditions (Fig. 2b), which is in line with previous studies (e.g. Flin et al., 2004; Calonne et al., 2014; Schleef et al., 2014). The anisotropy factor $Q$ (see Sect. 2.2) remained constant under isothermal conditions whereas temperature gradient metamorphism led to an increase of $Q$ between 50 h and 150 h due to the formation of ice structures that tend to be more aligned with the water vapor fluxes (Fig. 2c). This is in agreement with previous

5   studies (see e.g. Löwe et al., 2013; Calonne et al., 2014; Staron et al., 2014).

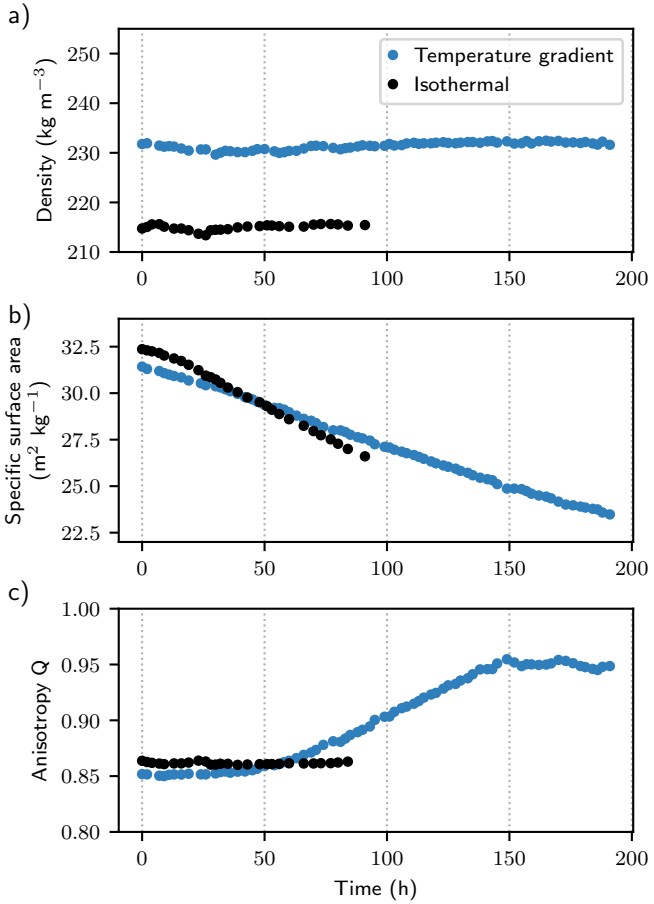

**Figure 2.** Time evolution of snow density (a), specific surface area (b) and anisotropy factor $Q$ (c) under the two types of metamorphism.

The dust volume estimated from the tomographic images was in average equal to $(2.3\pm0.05)\times10^{-9}$ m$^3$ ($\pm1\sigma$) in the temperature gradient experiment and $(2.6\pm0.08)\times10^{-9}$ m$^3$ in the isothermal experiment. The particles which can be detected both by tomography and Coulter analysis, only consists in the particles with a typical size in the range 5 to 20 $\mu$m. The particles within this size range represented a dust content of 0.2 mg g$^{-1}$ (40% of total) according to the Coulter analysis and more than

10   half of this content was detected by tomography. Almost none of the dust particles, mainly located in the middle of the sample, could escape the volume analyzed by image processing (see Sect. 2.2). The total dust content detected by tomography is

therefore expected to be constant with time. The relatively low standard deviation of the dust volume thus indicates that the detection of dust particles in the tomographic data was consistent.

## 3.2 Dust inclusion in the ice matrix

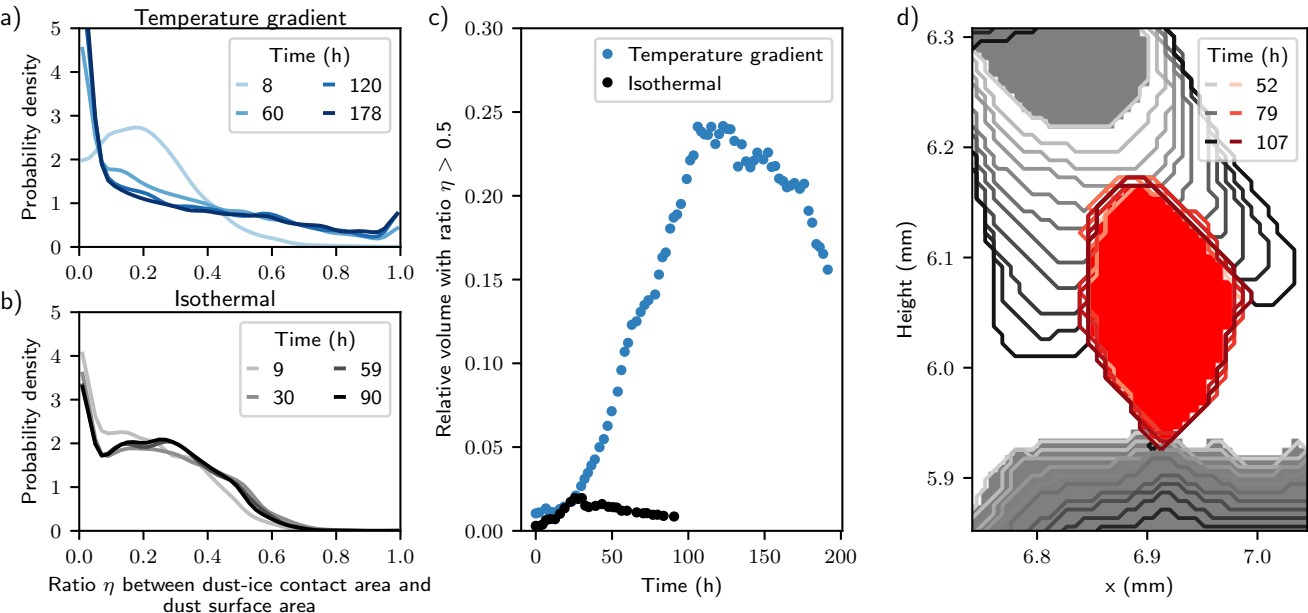

**Figure 3.** Dust inclusion in the ice matrix. (a, b) Distribution of the ratio $\eta$ between the dust-ice contact area and the dust surface area. The probability density is expressed in terms of number of particles and is computed with bins of width 0.02. (c) Time evolution of the relative volume of dust particles that share more than 50% of their surface with ice. (d) Example of the dust progressive inclusion in the ice matrix in the temperature gradient experiment. The initial location of ice (resp. dust) is shown with full gray (resp. red). Their time evolutions are shown by the different contours.

The relative position of dust particles in the ice matrix is affected by snow metamorphism. Some dust particles are obviously progressively embedded in the ice matrix during temperature gradient metamorphism (see supplementary Movie S1). To quantify this process, we computed the ratio $\eta$ between the dust-ice contact area and the dust surface area for each particle (see Sect. 2.2). At the beginning of the experiment, both samples exhibited similar distributions of $\eta$ characterized by low values. Under temperature gradient conditions, the number of dust particles with a high $\eta$ value clearly increased with time (Fig. 3a), while this trend was not present under isothermal conditions (Fig. 3b). The relative volume of dust that is more connected to ice than air ($\eta > 0.5$) increased from about 2% to 20% under temperature gradient conditions and remained stable around 2% under isothermal conditions (Fig. 3c). This volume decreased slowly under temperature gradient conditions after 100 h, which could correspond to the progressive release by the ice matrix of dust particles that were lying on the ice surface at the beginning of the experiment. We observed that the downward ice growth can completely surround dust particles from the top (Fig. 3d).

It is not the dust particle that enters the ice matrix but the ice that "absorbs" the particle. Note that our observations may be impacted by the nature of the LAP, e.g. by its conductivity or water affinity.

## 3.3 Downward movement of dust particles

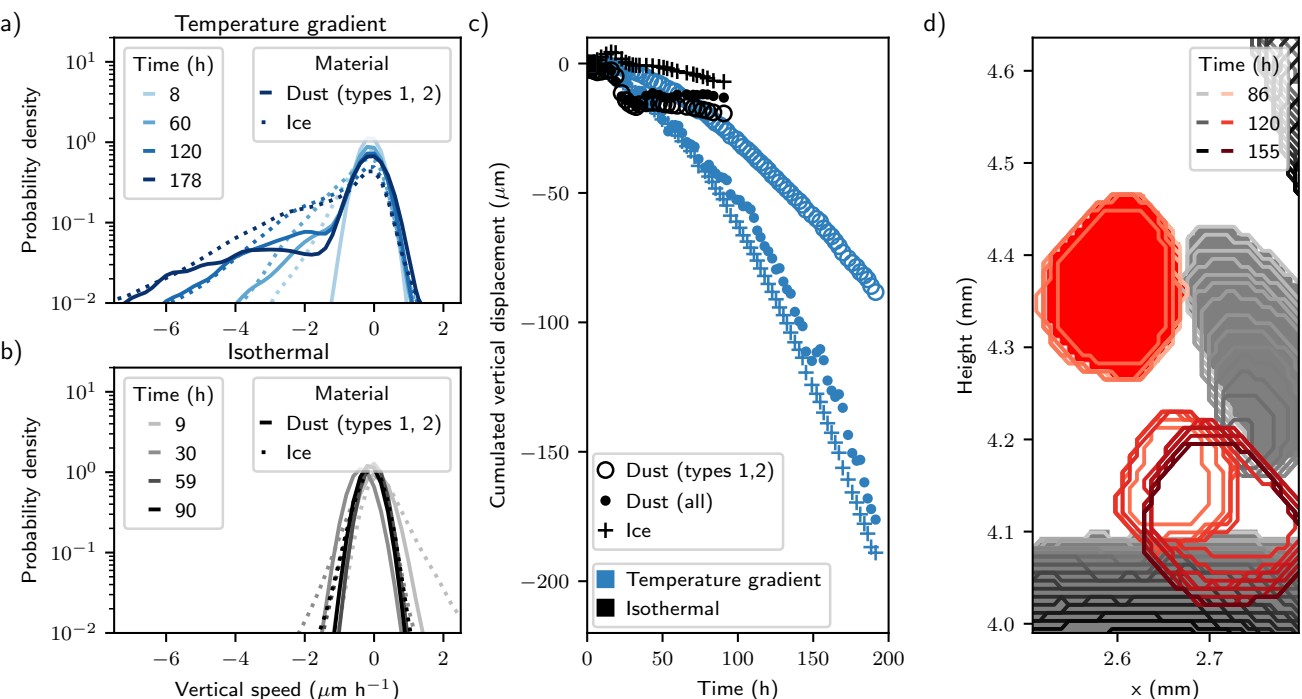

**Figure 4.** Dust downward movement. (a, b) Distribution of the dust particle and ice surface vertical velocities. The probability density is expressed in terms of number of particles and is computed with bins of width 0.02 $\mu$m h$^{-1}$. (c) Mean vertical velocity of the ice surface and of dust computed with all dust particles or only those with moderate displacements (types 1 and 2). (d) Example of a particle movement of type 3: sudden fall by loss of contact to ice and then progressive downward movement with the sublimating ice surface.

The ice interface is growing in the direction of the temperature gradient, while the vapor flux is in the opposite direction, which results in an overall positive upward water mass flux (e.g. Yosida, 1955; Flin and Brzoska, 2008; Pinzer et al., 2012). In contrast, dust does not sublimate and is thus necessarily characterized by a downward movement driven by gravity. We classified three main types of dust movement by order of displacement magnitude and continuity in time: (type 1) small movements due to the gravity-driven settlement of the ice matrix, (type 2) moderate movements of dust lying on a sublimating ice surface and (type 3) large and sudden movements due to particles falling into the pore space after loss of contact with the ice matrix.

Three different techniques were used to compute dust and ice movements from the tomographic images (see Sect. 2.2). The apparent movement of the ice structure is mainly due to sublimation-condensation at its surface and possibly slight gravity-

driven settlement of its volume. This movement was characterized by the displacement of the ice surface via a levelset approach. In contrast, dust particles move as solid bodies. The overall dust movement was characterized by the displacement of its center of mass. The individual movement of each particle was computed by digital image correlation. The latter technique does not capture large movements and we can thus assume reasonably that only movements of types 1 and 2 are captured by this method.

Figures 4a and b show the distribution of dust and ice surface vertical speeds for the two samples. The movements in the isothermal experiment (typically less than 4 $\mu$m between two successive images) were on the order of magnitude of the image analysis resolution. In contrast, non negligible movements of dust and ice were present in the temperature gradient experiment. The vertical speeds of ice and dust (not including type 3 movements) appear to be very close. Differences may occur by a preferential deposition of dust particles on flat upward-looking ice interfaces, that evolve slower than highly convex interfaces

aligned with the gradient.

      Figure 4c shows the cumulated displacement of the ice interface and of the dust particles. Again, vertical movements were only detectable for the temperature gradient experiment, for which a -200 $\mu$m mean displacement of the dust center of mass was observed. The mean displacement of all dust particles is very close to the one of the ice interface. However, even if the dust movement is driven by the ice movement and gravity, this resemblance seems coincidental since type 1 movements are

significantly smaller than the ice ones, and type 3 movements are significantly faster. Based on linear regressions between the cumulated displacements and time in the permanent regime (time larger than 140 h, Pearson correlation coefficient $R^2 > 0.98$), we estimated a total vertical speed of dust of -1.49 $\mu$m h$^{-1}$, with the falling dust (type 3) contributing to 55% of this movement. In addition, this global downward movement is accompanied by a vertical spread of the dust mass: +100 $\mu$m for the whole temperature gradient experiment, in contrast to +5 $\mu$m for the whole isothermal experiment.

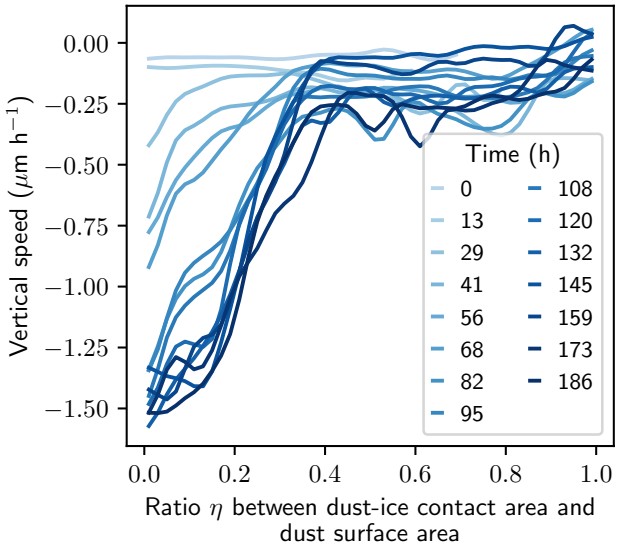

**Figure 5.** Link between dust movement (types 1, 2) and relative position to ice quantified by $\eta$ in the temperature gradient experiment.

The downward movement of dust is linked to its relative position to the ice matrix. Indeed, Fig. 5 shows that dust particles highly connected to the ice matrix (typically $\eta > 0.4$) do not move significantly. In contrast, particles slightly connected to the ice matrix exhibited larger mean downward movements. Particles embedded in the ice matrix can be considered as markers of the ice matrix settlement (type 1). Particles slightly connected to the ice matrix may follow the ice surface movement (type 2) or fall in the pore space (type 3). Indeed, a slight connection to the ice matrix implies that the "connecting" ice is also close to air and may therefore sublimate. In addition, the dust velocity varies with particles size: smaller particles are slightly faster than the larger ones for movement of type 1 and 2 (Fig. A3). This trend with particle size is important since smaller particles have more impact for albedo reduction (e.g. Flanner et al., 2012). It can be speculated that this trend is due to the fact that large particles have a larger absolute surface connected to the ice, thus reducing the speed of ice sublimation below the particle. For movements of type 3, we also expect the smaller particles to move faster than the large ones. Indeed, their fall into the pore space might be stopped later by the ice matrix, compared to the fall of larger particles (see Fig. 1). The distribution of $\eta$ did not exhibit any dependency to particle size (not shown).

## 3.4 Implications for arctic and sub-arctic snowpacks

The presented findings are especially important for arctic and sub-arctic snowpacks which often undergo strong temperature gradient metamorphism over several months. Domine et al. (2015) showed that in low-arctic shrub tundra (Quebec, Canada), the basal snow layer is exposed to a temperature gradient mostly above 20 K m$^{-1}$ between mid-November and early February. Assuming that the dust downward velocity remains constant and equal to -1.49 $\mu$m h$^{-1}$ during 3 months, or that the dust downward acceleration remains constant and equal to -0.0049 $\mu$m h$^{-2}$ (parabolic fit) during this period (Fig. 4c), the dust center of mass would move down by 3.2 mm or 22.9 mm, respectively. The particles would also spread vertically, as shown in the temperature gradient experiment by the increase of the standard deviation of dust position. Doherty et al. (2010) hypothesized that, for BC particles with a typical mass median diameter of 0.4 $\mu$m, such a downward movement would lead to a cleaning of the snow layer because the particles would end up on the ground. The vertical displacement measured in this study is rather small compared to the typical thickness of depth hoar layers (5-30 cm) found in arctic regions. However, the main driver of dust motion in dry snow is the presence of intensive sublimation and condensation. The movements of type 2 are directly linked to the local ice interface velocity and particles embedded within the ice matrix (type 1) would escape to enter movement of types 2 or 3 at a rate linked to the ice residence time (Pinzer et al., 2012). Therefore, dust motion may be faster for larger temperature gradients that are often found in arctic and sub-arctic snowpacks, e.g. roughly 50 K m$^{-1}$ on average over winter and up to 300 K m$^{-1}$, as reported by Sturm and Benson (1997). Moreover, small particles tend to move faster than the big ones (see Fig. A3), and particles whose size is lower than the image resolution (7.5 $\mu$m) were not detected here. We could thus hypothesize that BC particles would move downward faster than the dust particles analyzed here.

Eventually, we numerically evaluated the potential impact of the observed dust motion on the solar energy absorbed by the snowpack if the layer containing dust is at the snow surface. To this aim, we used the two-stream radiative transfer model TARTES (Libois et al., 2013). We considered a semi-infinite snowpack composed of dry snow with a density of 230 kg m$^{-3}$ and a specific surface area of 32 m$^2$ kg$^{-1}$, and four idealized hypothetical cases: (a) clean snowpack, (b) contamination with a

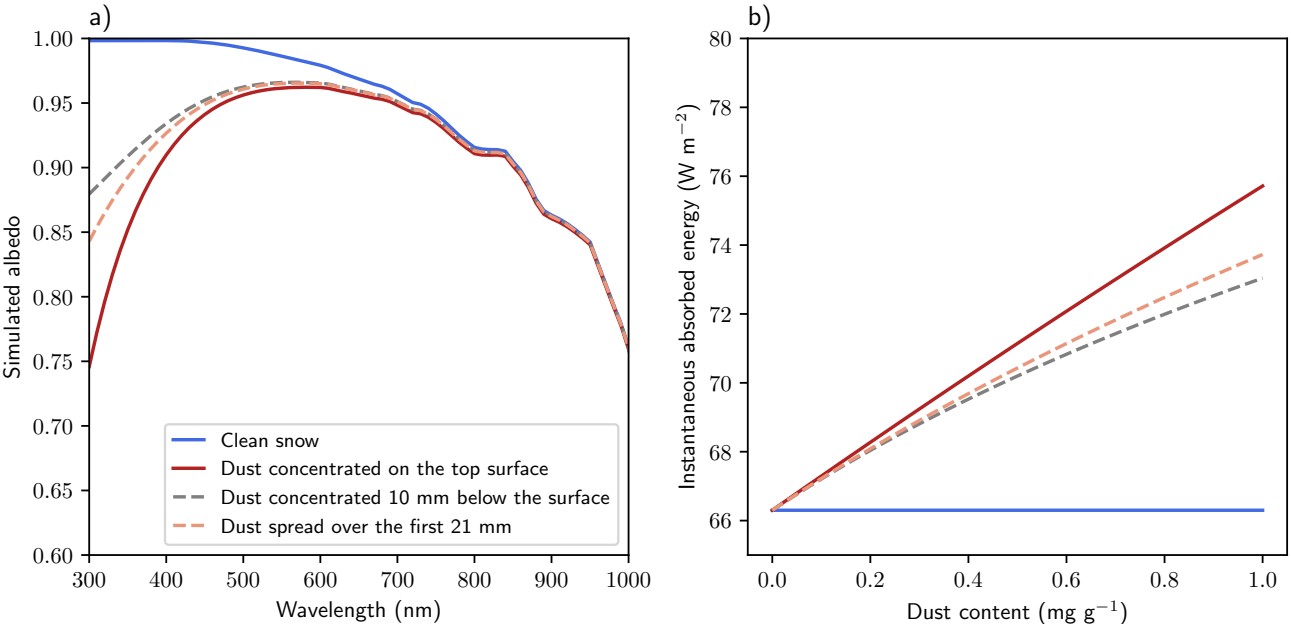

**Figure 6.** Simulated impact of dust motion on snow radiative properties. (a) Albedo as a function of wavelength for different idealized snowpacks and an initial dust concentration of 1 mg g$^{-1}$ (b) Instantaneous absorbed energy at 70$^o$ solar zenith angle as a function of the initial dust content and for the same idealized snowpacks.

1 mm thick layer of dust (concentration of 1 mg g$^{-1}$) on top, (c) same contamination but the dust is moved 10 mm downwards and (d) same contamination but the amount of dust is spread over the top 21 mm. Case (d) roughly corresponds to the overall displacement extrapolated from our data and estimated for 3 months of temperature gradient (see calculation above). We assumed that the mixing state of dust in snow is the same in all cases, as it appears to stabilize after a few days (Fig. 3). The
dust particles are all assumed to be outside the ice matrix (external mixture). We considered typical mass absorption efficiency of mineral dust as measured by Caponi et al. (2017) (sample Lybia PM_2.5 therein) for the mass fraction of dust particles of aerodynamic diameter lower than 2.5 $\mu$m. The simulated particles are thus smaller than those followed with the tomographic images. This choice was made to be consistent with the common atmospheric dust found in snow (e.g. Hess et al., 1998). The impact on albedo simulated here will also vary significantly for other types of dust with different chemical composition (e.g.
Table 4 in Caponi et al. (2017)). The albedo in the visible wavelengths is affected by the presence and location of the dust (Fig. 6a). Exposed to a typical winter incoming solar radiation with a solar zenith angle of 70$^o$ (instantaneous incoming energy assumed to be 376 W m$^{-2}$), the instantaneous energy absorbed by the snowpack is 66, 76, 73 and 74 W m$^{-2}$ respectively to cases a, b, c and d (Fig. 6a). Figure 6b displays the instantaneous absorbed energy for this incoming radiation, the four idealized snowpacks and varying dust content. It shows that when dust content reaches 0.5 mg g$^{-1}$, differences larger than 1 W m$^{-2}$ can
be found between the different dust locations. Note that this impact remains relatively small compared to the one due to dust

burial by new snow deposition, which may episodically occur during the winter season. Moreover, Aoki et al. (2014) showed that the sublimation of the snow surface to the atmosphere can enhance the concentration of LAPs at the topmost layer, which may counterbalance the optical impact of the observed downward movement.

## 4 Conclusions

This study presents the first *in operando* microscopic observation of the interactions between dust and snow metamorphism. We showed that temperature gradient metamorphism impacts LAP location at microscopic and macroscopic scales. Indeed, it affects the location of the particles with respect to the ice matrix and their relative vertical position in the snow layer. In the temperature gradient experiment, we observed a progressive embedding of dust particles within the ice matrix and a downward displacement of the dust center of mass by 200 $\mu$m on average for 200 hours. The dust motion mainly results from the fall of the dust particles into the pore space, while embedded particles only move with the settlement of the ice matrix. The comparison with the isothermal experiment, where no significant motion was observed, confirms that the main driver of dust motion in dry snow is the presence of intensive water vapor fluxes related to sublimation-condensation mechanisms. Such data are crucial for our understanding of LAP interactions with snow and the associated modification of the radiation balance. In typical arctic and sub-arctic conditions where strong temperature gradient metamorphism prevails, we estimated that dust motion could lead to a decrease of the solar energy absorbed when the dust is located on the top layer of the snowpack for a long duration. This study has implications for the evolution of snow physical and chemical properties on the ground and its impacts, e.g. water availability and avalanche hazard. Besides, this time series of high spatial and temporal resolution could benefit more broadly to studies focusing on the time evolution of the snow microstructure. Indeed, computing the ice surface velocity is notoriously difficult (e.g. Krol and Löwe, 2016) due to discretization artifacts and the unavoidable assumption that the velocity vector is normal to the ice surface. The acquired data might help overcome such difficulties.

The observation of the motion of LAPs in snow is here based on only one experiment of each temperature regime (isothermal and steady state temperature gradient), using mineral dust (Mongolian sand) and samples of recent snow confined between two ice lenses. Further experimental testing would help to confirm the presented results but also to assess the impact of the temperature gradient conditions (e.g. mean temperature, gradient magnitude, alternating sign) and initial snow microstructure on the dust speed. In addition, it would be interesting to study (1) whether the chemical composition and size of the LAPs impact the observed motion (e.g. for BC), (2) whether temperature gradient metamorphism on longer periods could lead to complete snow cleaning (Doherty et al., 2010) and (3) whether the observed motion can be counter-balanced by snow sublimation into the atmosphere, process by which the dust tends to concentrate at the snow surface (Aoki et al., 2014).

*Data availability.* The data presented in this paper are available under DOI: https://doi.pangaea.de/10.1594/PANGAEA.904568. The image processing library SimpleITK is available online at: http://www.simpleitk.org/. The radiative transfer model TARTES is available online at:

*Video supplement.* Two movies are provided as supplementary material:

– File S1.mp4: Movie of the time evolution of the snow microstructure and dust location under temperature gradient metamorphism. The snow is shown in gray, the dust is shown in red. The sample width is 10 mm. For visualization purposes, only a vertical slice of 0.375 mm thickness is shown.

– File S2.mp4: Movie of the time evolution of the snow microstructure and dust location under isothermal metamorphism. The snow is shown in gray, the dust is shown in red. The sample width is 10 mm. For visualization purposes, only a vertical slice of 0.375 mm thickness is shown.

# Appendix A

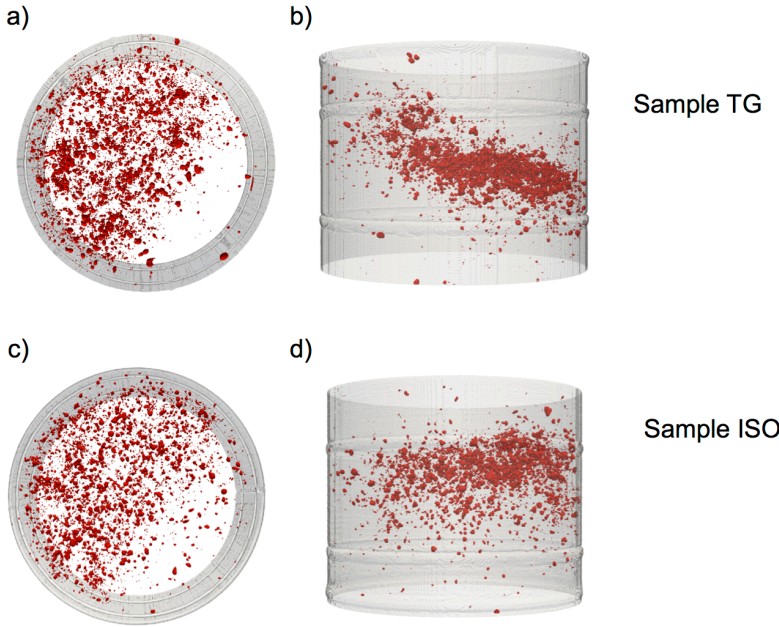

**Figure A1.** 3D view of the initial dust location (red) in the temperature gradient experiment (a, b) and the isothermal experiment (c, d) from top view (a, c) or side view (b, d). The aluminum sample holder inner contour is shown in transparent gray. The scale is indicated by the sample holder inner diameter, which is 10 mm.

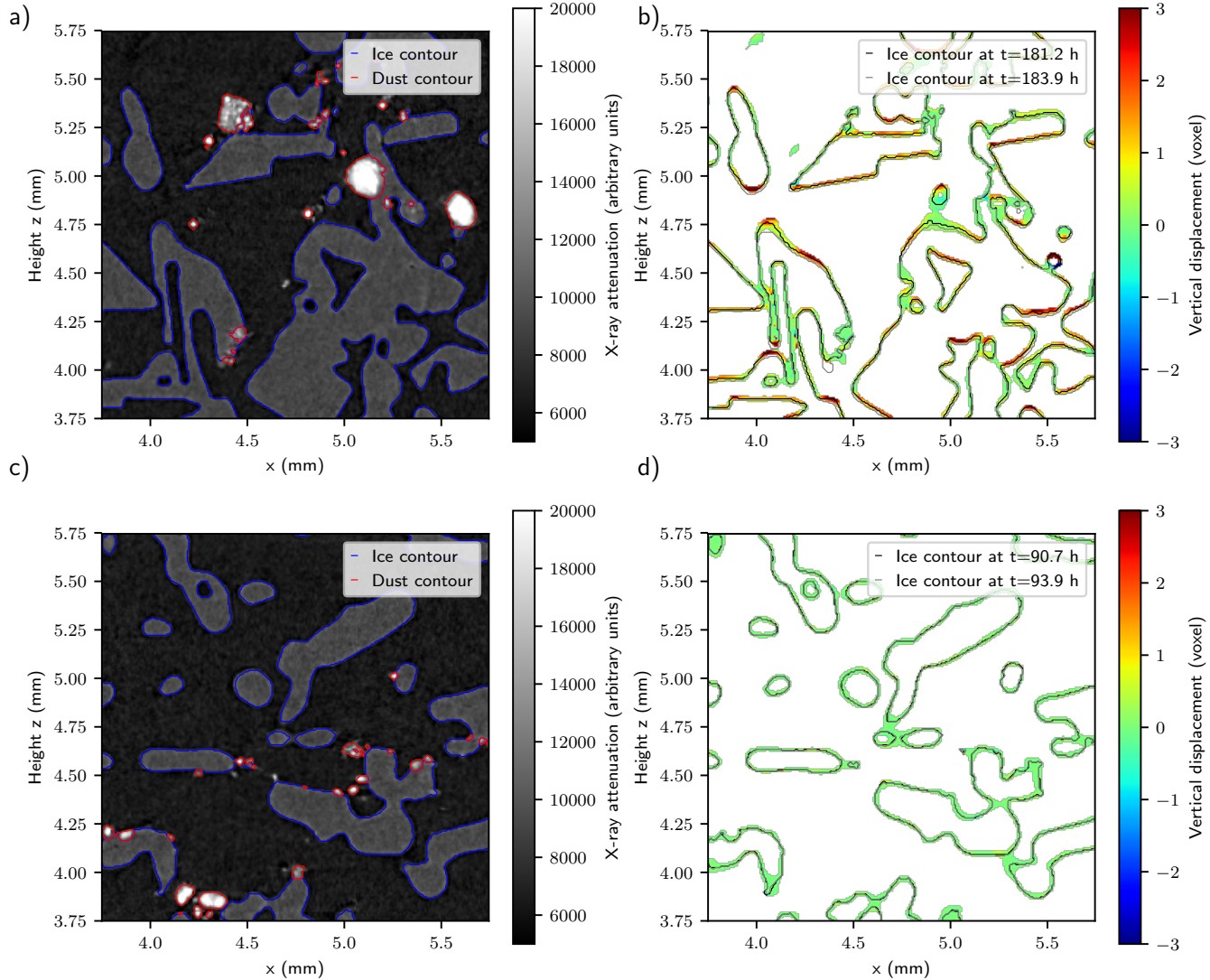

**Figure A2.** Example of the image processing workflow in the temperature gradient experiment (a, b) and the isothermal experiment (c, d): binary segmentation of the grayscale X-ray attenuation image (a, c), computation of ice interface velocity using the levelset approach (b, d). Note that for visualization purposes, the ice interface velocity is shown on a thicker interface (8 voxels width) compared to the computation used for the analysis (2 voxels). The vertical displacement between two images is shown. For example, a value of -2 voxels means that the ice contour went down by 2 voxels (i.e. by 15 $\mu$m) between the two considered images.

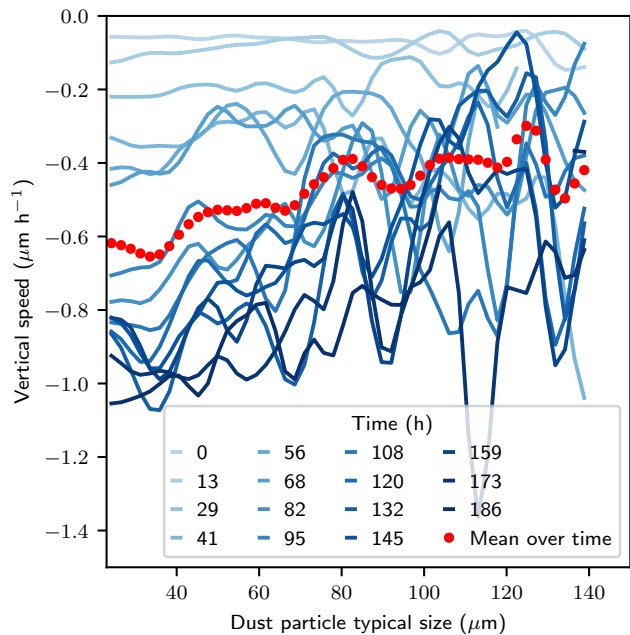

**Figure A3.** Mean vertical speed as a function of dust particle size for the temperature gradient experiment. Note that, here, only movements of type 1 and 2 and particles of volume larger than $5^3$ voxels were taken into account.

*Author contributions.* The three main contributors are P.H., F.F. and M.D.. M.D. proposed and monitored the whole project. F.F. designed and coordinated the experiment. P.H. processed and analyzed the tomographic data. P.H. and M.D. wrote the manuscript. F.T., I.P., P.L., A.D., J.R., L.P. contributed to the experimental work. D.V. supervised the analysis of the dust content and its chemical composition. E.A., S.R. and P.C. provided valuable help for X-ray imaging.

5  *Competing interests.* The authors declare no competing financial interests.

*Acknowledgements.* CNRM/CEN and IGE are part of Labex OSUG@2020 (Investissements d'Avenir, grant ANR-10-LABX-0056). 3SR is part of Labex TEC21 (Investissements d'Avenir, grant ANR-11-LABX-0030). This work was funded by the French National Research Agency (ANR JCJC EBONI grant ANR-16-CE01-006). We thank F. Domine, Q. Krol, H. Löwe, S. Morin and S. Warren for fruitful discussions. We also acknowledge M. King and M. Lamare as dust providers; J.-P. Coutet, R. Grand, R. Granger and La Grave ski resort for
10  their help with snow collection; and C. Di Biagio for providing the elemental and mineralogical analysis.

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
