# Peer review of "Motion of dust particles in dry snow under temperature gradient metamorphism"

_The Cryosphere, 2019_

## Referee Comment (RC1) · Kevin Hammonds (Referee) · 17 Apr 2019

In this paper, the authors provide both quantitative and qualitative evidence in support of their findings, that dust particles in dry snow can migrate within a snowpack when placed under an imposed temperature gradient. These experiments were conducted from within a controlled laboratory setting, such that the imposed temperature gradient could be finely controlled (18K/m), and also compared to a parallel experiment conducted under near-isothermal (< 5K/m) conditions. The light absorbing particles (LAP's) were tracked with X-ray computed microtomography (micro-CT) in both experiments, and a migration rate was quantified, based on the imposed temperature gradient and center of mass of the LAP's. Three modes of migration were also identified. The authors then went on to demonstrate the impact of the spatial location of

the LAP's as a function of snow depth on the snow radiative properties. As cited by the authors, this work is of relevance to those studying 1) the microstructural evolution of a snowpack under natural conditions, 2) the effects of LAP's on the radiative energy balance and albedo of natural snowpacks, and 3) the optical properties of a snowpack under natural conditions.

Overall, this is a very well-written and organized paper, and in my opinion, due to the novelty of these experiments, the detail by which the authors analyzed their data in support of their findings, and the broader impacts on the cryospheric research community, I recommend this study for immediate publication in The Cryosphere, pending only minor technical/content revisions.

Recommendation: Publication with Minor Revisions

General Comments/Questions:

1) Because these experiments are worthy of being repeated and extended to other problems related to snow metamorphism and its radiative energy balance, it would be appreciated if you could provide some additional detail in the text related to the methodology of your laboratory experiments. (specific questions below)

Line 5, page 3: Exactly how was the top surface of your snow surface "contaminated with dust"? (i.e. sprinkling by hand, blowing, sieving, etc.?) Was this technique based on any previous study that could be cited?

Line 4-6, page 11: In these four idealized cases, how was the dust added in a 1 mm thick layer and how was it measured (case b) and how was it spread over the top 21 mm (case d)?

2) Previous micro-CT work on snow has often used a 0.5°-0.7° step size, a 180° rotation, and a 10-17 um resolution, thereby reducing the overall scan time of a similarly sized sample to ∼20-30 min (see Heggli et al 2009, Chen & Baker 2010, Wang & Baker 2013, Hammonds et al 2015, Weise & Schneebeli 2017, and others). In the work pre-

sented here, a finer temporal and spatial resolution was used (3 h scan increment, 7.5 um resolution, 360° rotation), but a ∼2 h scan time...Can you please comment on the necessity of this increased resolution compared to previous work and the use of a 360° rotation? Can you also comment about the potential for the inadvertent heating of the sample with such a long scan time? And whether or not an in situ temperature measurement was made in these experiments or others with such a long scanning period? Lastly, if this is the most spatially and temporally resolved micro-CT analysis of dry snow that has been performed (Line 29, page 3), which it appears to be, can you remark on the degree to which this study benefited from this increased resolution, compared to the established body of previous literature?

3) Regarding the three main types of dust movement, can you further explain your observation of Type 1 "creep of the ice matrix" (Line 7, page 7, Line 2, page 8, Line 10, page 9, Line 7, page 12)? More specifically, are you referring to the plastic deformation of individual ice grains and/or the ice matrix, or what some may consider "settling" within the test volume due to the gravitational force? If the latter, it is suggested that you change the text to reflect this, so as to not be confused with plastic deformation via creep (time dependent, low stress deformation). If the former, please provide additional detail as to how this deformation was measured and what kind of stress/load was causing this deformation.

4) Can you further elaborate on why "smaller particles are slightly faster than the larger ones"? (Line 12, page 9). Figure A.3 is not particularly convincing. Was this quantified over the entire size distribution of particles? Furthermore, what was the size distribution of the Mongolian Sand that was used?

5) Although the attention to detail demonstrated in this study is appreciated, it appears that repeating these experiments at least once could have provided a more robust set of conclusions with perhaps some margin of error that could then be included for the benefit of other researchers in the cryospheric sciences...can you please comment on why only one experiment of each temperature regime was conducted? And whether or

not you recommend further laboratory testing?

References:

Chen, S., and I. Baker (2010), Evolution of individual snowflakes during metamorphism, Journal of Geophysical Research: Atmospheres, 115(D21).

Hammonds, K., R. Lieb-Lappen, I. Baker, and X. Wang (2015), Investigating the thermophysical properties of the ice–snow interface under a controlled temperature gradient, Cold Regions Science and Technology, 120, 157-167, doi:10.1016/j.coldregions.2015.09.006.

Heggli, M., E. Frei, and M. Schneebeli (2009), Snow replica method for three-dimensional X-ray microtomographic imaging, Journal of Glaciology, 55(192), 631-639.

Wang, X., and I. Baker (2013), Observation of the microstructural evolution of snow under uniaxial compression using X-ray computed microtomography, Journal of Geophysical Research: Atmospheres, 118(22), 12,371-312,382, doi:10.1002/2013jd020352.

Wiese, M., and M. Schneebeli (2017), Snowbreeder 5: a Micro-CT device for measuring the snow-microstructure evolution under the simultaneous influence of a temperature gradient and compaction, Journal of Glaciology, 63(238), 355-360.

---

## Referee Comment (RC2) · Teruo Aoki (Referee) · 22 Apr 2019

This paper describes the result of laboratory experiment using X-ray tomography on movement of dust particles in dry snowpack under temperature gradient metamorphism and isothermal metamorphism conditions. The paper clearly shows the dust particles move downward in case of temperature gradient metamorphism, in which three types of motion mechanisms are confirmed. They also present the quantitative vertical speed of the movement depending on relative position of dust particles to the ice matrix and estimated the total possible displacement of the dust distribution over dry snow period in the arctic. The authors discuss the influence of motion of light absorbing particles (LAPs) under strong temperature gradient in the arctic snow and the potential impact of dust vertical distribution near the snow surface on the radiative

properties.

The manuscript is well-written with the effective presentations including movies. This study gives new findings on the time evolution of vertical distribution of LAIs in snowpack which is valuable information to communities of snow microphysics and climate studies. I recommend this paper for publication in The Cryosphere after revising the following points:

Specific comments: p.2, L9-11: "Typically, the albedo decreases more when the mass of LAPs is concentrated in the first centimeter compared to the case when the mass is distributed over several centimeters (Dumont et al., 2014)." The effect of the vertical inhomogeneity of snow impurities was already investigated by Aoki et al. (2000), in which the same situation is represented.

p.9, L12-13: "In addition, the dust velocity varies with particles size: smaller particles are slightly faster than the larger ones (Fig. A3)." Please describe the (possible) cause. This is because the smaller dust particles as well as BC are more important for the albedo reduction. We should understand this mechanism.

p.11, L1ff and Figure 6: (1) Figure 6 suggests possible albedo increase due to the downward displacement of dust particles near the snow surface. Water vapor sometimes sublimates to the atmosphere from snow surface, which enhances the concentration of LAPs at the topmost layer (Aoki et al., 2014). This is opposite effect for the albedo change discussed here. Please describe that motion of dust particles near the surface could be affected by the other factor such as sublimation from snow surface.

(2) Albedo reduction due to dust contamination in snow depends on size distribution of dust particles. If the authors assume it based on the dust particles used in this experiment (i.e., Mongolian sand), it would be larger than the common atmospheric dust (e. g., A mode radius of dust model "Mineral-transported" compiled by Hess et al. (1998) is 0.5 $\mu$m.) and thus the estimated albedo reduction could be smaller than usual. Please indicate the size distribution parameters or single scattering parameters

of dust particles employed here.

(3) When upper part of snowpack is heated by solar radiation in daytime, the temperature gradient would be inverse near the surface (Pinzer and Schneebeli, 2009). In that case the vertical movement speed of dust particles due to temperature gradient metamorphism may differ from the result presented in this paper (e. g., the speed slows down?). Please mention on this situation briefly.

Figures 5-6 and A2-3: Some label values of both X and Y-axes are unreadable characters.

References: Aoki, T., T. Aoki, M. Fukabori, A. Hachikubo, Y. Tachibana, and F. Nishio (2000), Effects of snow physical parameters on spectral albedo and bidirectional reflectance of snow surface, J. Geophys. Res., 105(D8), 10,219–10,236, doi:10.1029/1999JD901122.

Aoki, T., Matoba, S., Yamaguchi, S., Tanikawa, T., Niwano, M., Kuchiki, K., Adachi, K., Uetake, J., Motoyama, H., and Hori, M., (2014), Light-absorbing snow impurity concentrations measured on Northwest Greenland ice sheet in 2011 and 2012, Bull. Glaciol. Res., 32, 21–31, doi:10.5331/bgr.32.21.

Hess, M., P. Koepke, and I. Schult, (1998), Optical properties of aerosols and clouds: The software package OPAC, Bull. Am. Meteorol. Soc., 79, 831-844, doi:10.1175/1520-0477(1998)079<0831:OPOAAC>2.0.CO;2.

Pinzer, B. R., and M. Schneebeli (2009), Snow metamorphism under alternating temperature gradients: Morphology and recrystallization in surface snow, Geophys. Res. Lett., 36, L23503, doi:10.1029/2009GL039618.

---

## Author Comment (AC1) · 14 Jun 2019

Answer to Kevin Hammonds (Referee):

We would like to thank Kevin Hammonds for this thorough analysis of our work and positive and constructive feedback, which helped us to improve the paper. The reviewer initial comments are written in black, our answer in blue and the corrections in the paper are highlighted in red. The line numbers used in the answers correspond to those of the corrected paper version.

In this paper, the authors provide both quantitative and qualitative evidence in support of their findings, that dust particles in dry snow can migrate within a snowpack when placed under an imposed temperature gradient. These experiments were conducted from within a controlled laboratory setting, such that the imposed temperature gradient could be finely controlled (18K/m), and also compared to a parallel experiment conducted under near-isothermal (< 5K/m) conditions. The light absorbing particles (LAP's) were tracked with X-ray computed microtomography (micro-CT) in both experiments, and a migration rate was quantified, based on the imposed temperature gradient and center of mass of the LAP's. Three modes of migration were also identified. The authors then went on to demonstrate the impact of the spatial location of the LAP's as a function of snow depth on the snow radiative properties. As cited by the authors, this work is of relevance to those studying 1) the microstructural evolution of a snowpack under natural conditions, 2) the effects of LAP's on the radiative energy balance and albedo of natural snowpacks, and 3) the optical properties of a snowpack under natural conditions.

Overall, this is a very well-written and organized paper, and in my opinion, due to the novelty of these experiments, the detail by which the authors analyzed their data in support of their findings, and the broader impacts on the cryospheric research community, I recommend this study for immediate publication in The Cryosphere, pending only minor technical/content revisions.

Recommendation: Publication with Minor Revisions

General Comments/Questions:
1) Because these experiments are worthy of being repeated and extended to other problems related to snow metamorphism and its radiative energy balance, it would be appreciated if you could provide some additional detail in the text related to the methodology of your laboratory experiments. (specific questions below)

Line 5, page 3: Exactly how was the top surface of your snow surface "contaminated with dust"? (i.e. sprinkling by hand, blowing, sieving, etc.?) Was this technique based on any previous study that could be cited?
The dust was sieved on the snow surface with a textile mesh (see Figure below). Unfortunately, to our knowledge, this does not correspond to a previous study that could be cited. Details on the procedure are now added in the text l. 6-8 p. 3: "Half of its top surface was contaminated by manually sieving dust with a metallic sieve ($0.5~x~0.5~mm^2$ holes). To smoothen this process, a textile mesh ($0.5~x~1~mm^2$ holes) has been previously folded and placed in the sieve before operation. Another snow layer of same properties was then sieved on top of the whole surface."

[Figure]

**Figure 1:** Sieve with textile mesh used to spread dust on the snow surface (the spoon gives the scale)

Line 4-6, page 11: In these four idealized cases, how was the dust added in a 1 mm thick layer and how was it measured (case b) and how was it spread over the top 21 mm (case d)?
The four idealized cases are only numerical (input of numerical model TARTES) and do not correspond to real physical experiments. Case (b) corresponds to a semi-infinite clean snowpack with, on top, 1 mm thick layer of snow homogeneously contaminated with dust at a concentration of 1 mg/g. Case (d) corresponds to the same amount of dust but homogeneously spread on the top 21 mm layer. These numerical experiments were conducted in order to mimic the overall displacement estimated in section 3.4.
To avoid this misunderstanding, we now clarified the sentence (l.31, p.11 to l.3 p.12): "Eventually, we numerically evaluated the potential impact of the observed dust motion on the solar energy absorbed by the snowpack if the layer containing dust is at the snow surface (…) We considered (…) four idealized hypothetical cases ...21 mm. Case (d) roughly corresponds to the overall displacement extrapolated from our data and estimated for 3 months of temperature gradient (see calculation above)."

2) Previous micro-CT work on snow has often used a 0.5°-0.7° step size, a 180° rotation, and a 10-17 um resolution, thereby reducing the overall scan time of a similarly sized sample to   20-30 min (see Heggli et al 2009, Chen & Baker 2010, Wang & Baker 2013, Hammonds et al 2015, Wiese & Schneebeli 2017, and others). In the work presented here, a finer temporal and spatial resolution was used (3 h scan increment, 7.5 um resolution, 360° rotation), but a   2 h scan time … Can you please comment on the necessity of this increased resolution compared to previous work and the use of a 360°rotation?
We used the highest resolution (7.5 microns/pixel) of the available tomographic setup to capture as much as dust particles as possible. Indeed, a post-mortem Coulter analysis showed that the mean mass diameter was about 5 microns for dust particles with a diameter between 0.6 and 20 microns. We would thus have missed a significant amount of dust if the scans were conducted with a rougher resolution (e.g. 36 microns in Wiese and Schneebeli, 2017). At this high resolution, the power of the incoming X-ray must be limited (here 7.5 W) to maintain a small size of the X-ray emission point, which consequently requires a smaller frame rate and thus lengthens the scanning time. In addition, as a rule of thumb, the number of projections must be of the order of the image width, to get an effective resolution close to the pixel size. It is not strictly equivalent to (1) measure 1440 projections on 180° (+ 2 times the beam angle) or to (2) measure the same number of projections on 360°. The method (2) generally provides 3D reconstructions with less artifacts compared to (1) (see e.g. Goyens et al. (2017), or Figure 2 below). Besides, the time evolution of the snow microstructure is sufficiently slow (see Fig. 5a or Fig. A2b of the article) to be correctly captured by scan at 7.5 microns lasting 2 hours and measured every 3 hours. This information is now added explicitly in the text as (l. 33-35 p.3) "The 3D image resolution (7.5 microns) and the subsequent tomograph settings were chosen to capture most of the dust particles with a mean mass diameter around 5 microns (according to the Coulter analysis for dust particles with a diameter between 0.6 and 20 microns) while keeping the scanning time (2 hours) short enough to correctly measure the microstructure time evolution."

[Figure]

**Figure 2:** Influence of the angular distribution of the projections on a reconstructed slice: (a) All 1440 projections on [0, 360] degree, (b) half (720) of the projections distributed on [0, 360] degree, (c) All (720) projections located between 0 and 180 degree.

Can you also comment about the potential for the inadvertent heating of the sample with such a long scan time? And whether or not an in situ temperature measurement was made in these experiments or others with such a long scanning period?
The sample temperature is controlled and monitored via the cryogenic cell CellDyM. Details on this control are now added l.18-21 p.3: "The temperature of the Peltier modules are monitored during the whole experiment and

controlled with a relative precision of about $\pm$0.01°C. The Pt100 probes were calibrated together in a thermo-regulated bath so that the accuracy of their temperature difference is below $\pm$0.02°C and their absolute temperature is known at $\pm$0.05°C. The insulation with vacuum is designed to avoid any inadvertent lateral heating of the sample (Calonne et al., 2015)".

Lastly, if this is the most spatially and temporally resolved micro-CT analysis of dry snow that has been performed (Line 29, page 3), which it appears to be, can you remark on the degree to which this study benefited from this increased resolution, compared to the established body of previous literature?

As explained just before (and now mentioned in the article), this high resolution was required to capture most of the dust particles. Without focusing on the application to dust analysis, this time series of high spatial and temporal resolution could benefit to studies focusing on the time evolution of the snow microstructure. Indeed, computing the ice surface velocity is notoriously difficult (e.g. Krol and Löwe, 2016) due to discretization artifacts and the unavoidable assumption that the velocity vector is normal to the ice surface. The acquired data would help overcome such difficulties. This information is now added in the conclusion l.17-20 p.13: "Besides, without focusing on the application to dust analysis, this time series of high spatial and temporal resolution could benefit more broadly to studies focusing on the time evolution of the snow microstructure. Indeed, computing the ice surface velocity is notoriously difficult (e.g. Krol and Löwe, 2016) due to discretization artifacts and the unavoidable assumption that the velocity vector is normal to the ice surface. The acquired data might help overcome such difficulties."

3) Regarding the three main types of dust movement, can you further explain your observation of Type 1 "creep of the ice matrix" (Line 7, page 7, Line 2, page 8, Line 10, page 9, Line 7, page 12)? More specifically, are you referring to the plastic deformation of individual ice grains and/or the ice matrix, or what some may consider "settling" within the test volume due to the gravitational force? If the latter, it is suggested that you change the text to reflect this, so as to not be confused with plastic deformation via creep (time dependent, low stress deformation). If the former, please provide additional detail as to how this deformation was measured and what kind of stress/load was causing this deformation.

The term we used might effectively be confusing. We meant slow deformation of the ice matrix due to gravitational force, which might reflect the plastic deformation of individual ice grains but we cannot assess whether this overall deformation only consists of creep and not other deformation processes (e.g. viscosity). To avoid this confusion, we replaced "ice creep" by "ice matrix settling" everywhere in the text.

4) Can you further elaborate on why "smaller particles are slightly faster than the larger ones"? (Line 12, page 9). Figure A.3 is not particularly convincing. Was this quantified over the entire size distribution of particles? Furthermore, what was the size distribution of the Mongolian Sand that was used?

The speed and surface areas of individual grains were computed only for dust particles with a volume larger than 5**3 voxels and for dust movements of type 1 and 2. This information was not clear from the text and is now added as (l. 19-21, p. 5): "We rejected the displacements when the tracking metric, namely normalized cross correlation, was lower than 0.9 or when they were computed on a too small volume (dust particles volume lower than 5**3 voxels)." That is why Figure A3 only shows the mean speed of particles with a typical size larger than (3V/4pi)**(1/3) =23 microns. The overall size distribution of the particles detected by tomography is shown below:

[Figure]

**Figure 3:** Distribution of dust particle typical size, as computed from tomographic data. The probability distribution is expressed in terms of relative volume.

To make Figure A3 clearer, we add the mean vertical speed averaged on time on Figure A3. A decreasing trend of absolute particle speed with size can now be clearly seen. Moreover, we now also speculate on the origin of this trend l. 8-11, p. 11: "It can be speculated that this trend is due to the fact that large particles have a larger absolute surface connected to the ice, thus reducing the speed of ice sublimation below the particle. For movements of type 3, we also expect the smaller particles to move faster than the large ones. Indeed, their fall into the pore space might be stopped later by the ice matrix, compared to the fall of larger particles (see Fig. 1)."

[Figure]

**New figure A3:** Mean vertical speed as a function of dust particle size for the temperature gradient experiment. Note that here only movements of type 1 and 2 and particles of volume larger than 5**3 voxels were taken into account.

5) Although the attention to detail demonstrated in this study is appreciated, it appears that repeating these experiments at least once could have provided a more robust set of conclusions with perhaps some margin of error that could then be included for the benefit of other researchers in the cryospheric sciences… can you please comment on why only one experiment of each temperature regime was conducted? And whether o not you recommend further laboratory testing?

Only one experiment of each temperature regime was conducted because of the rather complex sample preparation, the important human (almost permanent human supervision of the experiment) and material resources (> 500€/day of tomography) required to run the tomographic acquisition. Newly acquired tomographic instrument (TomoCold) dedicated to snow and ice studies and located in a cold room might help to overcome these limitations.

We agree with the reviewer and feel that additional experiments would help to further elucidate the following questions:
- Does the chemical composition of the LAPs affect the observed behavior. For instance, would BC behave the same way as mineral dust?
- When does the observed dust motion reach a permanent regime (constant downward speed)? Can it cause complete snow cleaning?
- How does the interaction with the atmosphere affect the observed behavior? Here, the sample was confined in a certain sample holder and the snow surface was not exposed to opened atmosphere and sunlight. In natural conditions and under temperature gradient, snow at the surface also sublimates into the atmosphere, which may concentrate dust at the snow surface and cause an upward movement of the particles with respect to the snow surface (Aoki et al., 2014). Moreover, direct radiative impact on the dust particles may also affect their location.
- How do the temperature gradient characteristics (mean T, magnitude, sign) affect dust motion?

The following paragraph was added at the end of the conclusion (l. 21-28, p. 13): "The observation of the motion of LAPs in snow is based on only one experiment of each temperature regime (isothermal and steady state temperature gradient), using mineral dust (Mongolian sand) and snow samples confined between two ice lenses. Further experimental testing would help to confirm the presented results but also to assess the impact of the temperature gradient regime (e.g. mean temperature, gradient magnitude, alternating sign) and initial snow microstructure on the dust speed. In addition, it would be interesting to study (1) whether the chemical composition and size of the LAPs impact the observed motion (e.g. for BC), (2) whether temperature gradient metamorphism on longer periods could lead to complete snow cleaning (Doherty et al., 2010) and (3) whether the observed motion can be counter-balanced by snow sublimation into the atmosphere which tends to concentrate dust at the snow surface (Aoki et al., 2014)."

References (only new ones):

Chen, S., and I. Baker (2010), Evolution of individual snowflakes during metamorphism, Journal of Geophysical Research: Atmospheres, 115(D21).

Hammonds, K., R. Lieb-Lappen, I. Baker, and X. Wang (2015), Investigating the thermophysical properties of the ice–snow interface under a controlled temperature gradient, Cold Regions Science and Technology, 120, 157-167, doi:10.1016/j.coldregions.2015.09.006.

Heggli, M., E. Frei, and M. Schneebeli (2009), Snow replica method for three- dimensional X-ray microtomographic imaging, Journal of Glaciology, 55(192), 631-639.

Wang, X., and I. Baker (2013), Observation of the microstructural evolution of snow under uniaxial compression using X-ray computed microtomography, Journal of Geophysical Research: Atmospheres, 118(22), 12,371-312,382, doi:10.1002/2013jd020352.

Wiese, M., and M. Schneebeli (2017), Snowbreeder 5: a Micro-CT device for measuring the snow-microstructure evolution under the simultaneous influence of a temperature gradient and compaction, Journal of Glaciology, 63(238), 355-360

Aoki, T., Matoba, S., Yamaguchi, S., Tanikawa, T., Niwano, M., Kuchiki, K., Adachi, K., Uetake, J., Motoyama, H., and Hori, M., (2014), Light-absorbing snow impurity concentrations measured on Northwest Greenland ice sheet in 2011 and 2012, Bull. Glaciol. Res., 32, 21–31, doi:10.5331/bgr.32.21.

Goyens, J., Mancini, L., Van Nieuwenhove, V., Sijbers, J., & Aerts, (2017), P. Comparison of conventional and synchrotron X-ray microCT scanning of thin membranes in the inner ear. Micro-CT user Meeting 2017 (Bruker), Brussels, 12-15 June 2017.

Krol Q. and Löwe H. (2016) Analysis of local ice crystal growth in snow. *Journal of Glaciology* **62**(232), 378–390 (doi:10.1017/jog.2016.32)

---

## Author Comment (AC2) · 14 Jun 2019

This paper describes the result of laboratory experiment using X-ray tomography on movement of dust particles in dry snowpack under temperature gradient metamorphism and isothermal metamorphism conditions. The paper clearly shows the dust particles move downward in case of temperature gradient metamorphism, in which three types of motion mechanisms are confirmed. They also present the quantitative vertical speed of the movement depending on relative position of dust particles to the ice matrix and estimated the total possible displacement of the dust distribution over dry snow period in the arctic. The authors discuss the influence of motion of light absorbing particles (LAPs) under strong temperature gradient in the arctic snow and the potential impact of dust vertical distribution near the snow surface on the radiative properties.

The manuscript is well-written with the effective presentations including movies. This study gives new findings on the time evolution of vertical distribution of LAIs in snowpack which is valuable information to communities of snow microphysics and climate studies. I recommend this paper for publication in The Cryosphere after revising the following points:

We gratefully acknowledge Teruo Aoki for his profound analysis of our work and constructive feedback, which helped us to improve the paper. The reviewer initial comments are written in black, our answer in blue and the corrections in the paper are highlighted in red. The line numbers used in the answers correspond to those of the corrected paper version.

Specific comments:

p.2, L9-11: "Typically, the albedo decreases more when the mass of LAPs is concentrated in the first centimeter compared to the case when the mass is distributed over several centimeters (Dumont et al., 2014)." The effect of the vertical inhomogeneity of snow impurities was already investigated by Aoki et al. (2000), in which the same situation is represented.
We are sorry for this omission and thank the reviewer for this suggestion, which was added to the reference list (l. 11 p.2).

p.9, L12-13: "In addition, the dust velocity varies with particles size: smaller particles are slightly faster than the larger ones (Fig. A3)." Please describe the (possible) cause. This is because the smaller dust particles as well as BC are more important for the albedo reduction. We should understand this mechanism.
We agree that this phenomenon may have important implications for the resulting albedo and now mention it in the text. However, we feel that the presented data does not contain any clear evidence of the origin of the phenomenon. We now only speculate (l. 7-11, p. 11) that: "This trend with particle size is important since smaller particles have more impact for albedo reduction (Flanner et al., 2012). It can be speculated that this trend is due to the fact that large particles have a larger absolute surface connected to the ice, thus reducing the speed of ice sublimation below the particle. For movements of type 3, we also expect the smaller particles to move faster than the large ones. Indeed, their fall into the pore space might be stopped later by the ice matrix, compared to the fall of larger particles (see Fig. 1)."

p.11, L1ff and Figure 6:
(1) Figure 6 suggests possible albedo increase due to the downward displacement of dust particles near the snow surface. Water vapor sometimes sublimates to the atmosphere from snow surface, which enhances the concentration of LAPs at the topmost layer (Aoki et al., 2014). This is opposite effect for the albedo change discussed here. Please describe that motion of dust particles near the surface could be affected by the other factor such as sublimation from snow surface.
We fully agree with the reviewer. We now added this information in the text l. 1-3, p. 13: "Moreover, Aoki et al. (2014) showed that the sublimation of the snow surface to the atmosphere can enhance the concentration of LAPs at the topmost layer, which may counterbalance the optical impact of the observed downward movement" and in the perspectives l. 21-28, p. 13.

(2) Albedo reduction due to dust contamination in snow depends on size distribution of dust particles. If the authors assume it based on the dust particles used in this experiment (i.e., Mongolian sand), it would be larger than the common atmospheric dust (e. g., A mode radius of dust model "Mineral-transported" compiled by Hess et al. (1998) is 0.5 mum) and thus the estimated albedo reduction could be smaller than usual. Please indicate the size distribution parameters or single scattering parameters of dust particles employed here.

We used the mass absorption efficiency of mineral dust compiled by Caponi et al. (2017) and used the measurements obtained for sample "Lybia in the PM_2.5 size fractions" (Table 4, therein). This is now mentioned in the text l. 4-10, p. 12 as "We considered typical mass absorption efficiency of mineral dust as measured by Caponi et al. (2017) (sample Lybia PM_2.5 therein) for the mass fraction of dust particles of aerodynamic diameter lower than 2.5 microns. The simulated particles are thus smaller than the one followed with the tomographic images. This choice was made to be consistent with the common atmospheric dust found in snow, e.g. Hess et al. (1998). The impact on albedo simulated here will also vary significantly for other types of dust with different chemical composition (e.g. Table 4 in Caponi et al. 2017). "

(3) When upper part of snowpack is heated by solar radiation in daytime, the temperature gradient would be inverse near the surface (Pinzer and Schneebeli, 2009). In that case the vertical movement speed of dust particles due to temperature gradient metamorphism may differ from the result presented in this paper (e. g., the speed slows down?). Please mention on this situation briefly.

Measurements in Greenland and in alpine snowpacks have effectively shown that daily cycles of radiative heating and cooling may lead to an inversion of the temperature gradient in the topmost 20 cm. However, in arctic snowpacks, daily variations of temperature near the surface are less pronounced (Pinzer and Schneebeli, 2009). Pinzer and Schneebeli (2009) used temperature on the order of 100 K/m and observed large volumetric turnover of up to 60% of the ice mass in one half-cycle. In such a case, we would expect the observed motion to be of the same order as observed in this study since a large amount of the ice next to particles would sublimate and set the particles free into the pore space. With lower gradient magnitude, the particles motion may slow down with time since it tends to be always the "same ice" (the ice envelop below and above the grain static part), which is active (sublimates/condensates) and ends to be clean. This situation is now briefly mentioned in the perspectives as future work to be done:

"The observation of the motion of LAPs in snow is based on only one experiment of each temperature regime (isothermal and steady state temperature gradient), using mineral dust (Mongolian sand) and snow samples confined between two ice lenses. Further experimental testing would help to confirm the presented results but also to assess the impact of the temperature gradient regime (e.g. mean temperature, gradient magnitude, alternating sign) and initial snow microstructure on the dust speed. In addition, it would be interesting to study (1) whether the chemical composition and size of the LAPs impact the observed motion (e.g. for BC), (2) whether temperature gradient metamorphism on longer periods could lead to complete snow cleaning (Doherty et al., 2010) and (3) whether the observed motion can be counter-balanced by snow sublimation into the atmosphere which tends to concentrate dust at the snow surface (Aoki et al., 2014)."

Figures 5-6 and A2-3: Some label values of both X and Y-axes are unreadable characters.
We do not see this problem on our web browser with the online version. We will carefully check it with TC editing service.

References(only new ones):
Aoki, T., T. Aoki, M. Fukabori, A. Hachikubo, Y. Tachibana, and F. Nishio (2000), Effects of snow physical parameters on spectral albedo and bidirectional reflectance of snow surface, J. Geophys. Res., 105(D8), 10,219–10,236, doi:10.1029/1999JD901122.
Aoki, T., Matoba, S., Yamaguchi, S., Tanikawa, T., Niwano, M., Kuchiki, K., Adachi, K., Uetake, J., Motoyama, H., and Hori, M., (2014), Light-absorbing snow impurity concentrations measured on Northwest Greenland ice sheet in 2011 and 2012, Bull. Glaciol. Res., 32, 21–31, doi:10.5331/bgr.32.21.
Hess, M., P. Koepke, and I. Schult, (1998), Optical properties of aerosols and clouds: The software package OPAC, Bull. Am. Meteorol. Soc., 79, 831-844, doi:10.1175/1520-0477(1998)079<0831:OPOAAC>2.0.CO;2.
Pinzer, B. R., and M. Schneebeli (2009), Snow metamorphism under alternating temperature gradients: Morphology and recrystallization in surface snow, Geophys. Res. Lett., 36, L23503, doi:10.1029/2009GL039618
Flanner, M. G., Liu, X., Zhou, C., Penner, J. E., and Jiao, C.: Enhanced solar energy absorption by internally-mixed black carbon in snow grains, Atmos. Chem. Phys., 12, 4699-4721, https://doi.org/10.5194/acp-12-4699-2012, 2012.

---

## Author Response (AR2)

Dear Editor,

You commented about the paper tc-2019-41: "However, I could not find a data statement (for further advice see https://www.the-cryosphere.net/about/data_policy.html )- Please state in your manuscript where the data of the experiment are available, apologize if it is hidden in the text."

In the paragraph "Data Availability" l.29, p.13 (lines corresponding to the revised manuscript), it is mentioned that "The datasets analysed during the current study are available from the corresponding author on request." Is that not sufficient as a data statement or you just missed this sentence?

Best regards,

Pascal

---

## Author Response (AR3)

Dear Editor,

The data presented in the paper are now available under DOI :
https://doi.pangaea.de/10.1594/PANGAEA.904568, and the data availability section was changed accordingly as :

« The data presented in this paper are available under DOI:
https://doi.pangaea.de/10.1594/PANGAEA.904568. The image processing library SimpleITK is available online at: http://www.simpleitk.org/. The radiative transfer model TARTES is available online at: http://pp.ige-grenoble.fr/pageperso/picardgh/tartes/. Supporting 3D raw tomographic data can be obtained from the corresponding author upon request. »

Best regards,

Pascal